# Injury Prevention, Safety Education and Violence in Relation to the Risk of Tooth Fracture among Korean Adolescents

**DOI:** 10.3390/ijerph17228556

**Published:** 2020-11-18

**Authors:** Han-Na Kim, Yong-Bong Kwon, Min-Ji Byon, Jin-Bom Kim

**Affiliations:** 1Department of Dental Hygiene, College of Health and Medical Sciences, Cheongju University, Cheongju 28503, Korea; hnkim@cju.ac.kr; 2Department of Preventive and Community Dentistry, School of Dentistry, Pusan National University, Yangsan 50612, Korea; naeda1103@naver.com (Y.-B.K.); kyura2@naver.com (M.-J.B.); 3BK21 FOUR Project, Pusan National University, Yangsan 50612, Korea

**Keywords:** adolescent, injury prevention, oral health, safety education, tooth fracture

## Abstract

This study aimed to determine the injury prevention-, safety education-, and violence-related factors pertaining to tooth fracture experience (TFE) in Korean adolescents. We used data from the 14th Korea Youth Risk Behavior Survey (KYRBS) in 2018. The 60,040 participants were selected using a complex sampling design from middle and high schools. The participants completed a self-administered questionnaire. The explanatory variables, including school safety education and violence, were assessed in relation to prevention of traumatic injuries. Complex-samples multivariable logistic regression models were applied to explain the factors related to TFE over the past 12 months. The overall prevalence of TFE was 11.4%. Risk factors related to tooth fractures were not wearing a seatbelt on an express bus, not wearing helmets while riding motorcycles and bicycles, clinical treatment due to injuries at school, injuries associated with earphone/smartphones use, and lack of school safety education such as danger evacuation training. The top risk factor was injuries associated with earphone/smartphone usage, followed by lack of familiarity with school safety education. Thus, to prevent tooth fractures among adolescents, schools should strengthen their safety education, including education regarding mobile device usage, and wearing a seatbelt and wearing a helmet. Care should be taken to manage facilities around the school and to prevent injury related to tooth fracture. Further studies on various risk factors related to tooth fractures are warranted.

## 1. Introduction

The prevalence of traumatic dental injuries of primary and permanent dentition from a worldwide literature search was 22.7% and 15.2%, respectively [1]. According to the etiology of traumatic dental injuries in Brazil, 2003, tooth fracture experience (TFE) was greater in boys (13.6%) than in girls (7.6%) [2]. The burden of tooth fractures in Korean adolescents is high, with prevalence estimates ranging from 16.8% to 19.3% [3,4,5]. 

The most common dental injuries occur in the maxillary central incisors [6,7], which results in functional limitations in chewing and pronunciation [8,9]. In particular, tooth fractures in adolescents can affect their appearance and influence their daily life by affecting them psychologically and deteriorating their quality of life [10]. Studies related to TFE have reported not only the incidence rate according to sex [11,12,13], the location and type of injury [7,14,15,16], and the oral section [7,11,17,18], but also the environmental [19,20,21], socio-psychological and behavioral factors related to the socio-economic status of dental trauma patients [12,19]. However, comprehensive assessments of adolescents’ health behaviors and education for the prevention of TFE are relatively rare.

Adolescents spend a large proportion of their days in school or pursuing school-related activities. In school, adolescents receive conventional education as well as the experience and education to help them become healthy adults, in which education on health and safety is an essential component. Since its launch in 1995, the WHO’s Global School Health Initiative has sought to mobilize and strengthen school health programs globally. In effect, school health programs should strive to formulate health policies and provide safe and healthy environments, health education, and health services, including screening programs for various conditions and behaviors [22]. Compared to the school health program, adolescents had not received school safety education (SSE).

One previous study has reported that 70%–80% of adolescents do not wear safety tools, including helmets and seat belts [23]. Transport-related risk-taking behaviors, defined as passenger-, motorcycle-, and driving-related risk behaviors, account for the majority of adolescent injuries. In Australia, transport-related injuries accounted for 30% of all injury deaths among young people [24]. 

According to a study on smartphone dependence in 2017, the recent adolescence smartphone usage rates are 34.3% in middle school students and 28.7% in high school students, which are higher than that in university students (23.8%) and the highest rates among all age groups [25]. A study that identified the factors influencing the damage caused by smartphone usage reported a high probability of falling or bumping injuries when walking or performing other activities while using a smartphone [26]. Research on the experience of injuries in school has primarily focused on cases attributable to sexual violence, bullying, and stress [27]. However, usage of safety equipment and systematic safety education cannot completely prevent physical damage. Therefore, this study aimed to analyze protective or risk factors such as injury prevention, safety education, and violence in relation to tooth fractures among adolescents by using the 14th Korea Youth Risk Behavior Survey (KYRBS) data obtained from middle and high school students. The hypotheses of this study were as follows: (1) adolescents riding a motorcycle or bicycle without a helmet will have a higher TFE than those who use a helmet. (2) Adolescents who did not receive SSE will show a higher TFE than those who received SSE.

## 2. Materials and Methods 

### 2.1. Study Participants

This study used data from the 14th KYRBS conducted in 2018. This survey is an anonymous self-administered online survey that is conducted to analyze the health-risk behaviors among middle school and high school students in South Korea. These data were obtained from the official website of KYRBS from Korea Centers for Disease Control and Prevention (KCDC) (http://yhs.cdc.go.kr/) and downloaded after receiving approval from the relevant institution [28]. The survey, a government-approved survey conducted on the basis of Article 19 of the Law for the Promotion of the Nation’s Health, has been conducted annually since 2005 by the Ministry of Education, the Ministry of Health and Welfare, and KCDC [29].

The 14th KYRBS conducted in 2018 was a government-approved statistical survey (approval no. 117058) with a design aimed at representing Korean middle- and high-school students nationwide. In the sample distribution stage, the sample size was set to 400 middle schools and 400 high schools. The proportional distribution method was applied so that the population composition ratio and sample composition ratio for each stratification variable matched. Regional divisions were based on city, province, city scale (large city, small and medium city, and county area), and regional county. The number of sample schools was calculated according to the ratio of male and female students in middle schools and the ratio of male and female as well as educational characteristics in high schools. As a result, a total of 62,823 students from 400 middle schools and 400 high schools were designated as samples, and 30,463 male students and 29,577 female students participated in the actual survey. The participation rate was 95.6% based on the number of students [28]. The participants were aged 13–18 years-old. Sample participants from designated schools were randomly assigned to one computer per school computer room with internet access to answer the questionnaires. 

The questionnaires were developed through national and international data and expert advisory committees related to youth health. The survey contained a total of 103 questions covering 15 domains, namely smoking, drinking, physical activity, eating, obesity and weight control, mental health, prevention of injury, oral health, personal hygiene, sexual behavior, atopy and asthma, drugs, internet addiction, health equity, and violence [30]. 

### 2.2. Variables

Tooth fracture was included if the participants answered that they experienced symptoms of the tooth being fractured or broken, or that the tooth was fractured or broken due to movement or injury during the last 12 months. TFE was answered as a dependent variable. In this study, the relationships of the survey variables in the domains of injury prevention, safety education, and violence with TFE were selected as the analysis targets. 

Among variables in the injury prevention domain, the use of seatbelts in front or rear seats of cars, seatbelt wearing on an express bus, and helmet wearing while riding a motorcycle/bicycle in the last 12 months were selected as variables. In the SSE domain, experience of clinical injury treatment, experience of clinical injury treatment due to careless use of earphone/smartphone usage, and SSE for safety, injury prevention, evacuation, rescue and lifesaving, and cardiopulmonary resuscitation (CPR) were selected. The experience of treatment due to injury at school means that students have been injured in school (within school boundaries such as classrooms, hallways, playgrounds, etc.) and have been treated at a hospital. This refers to a situation caused by falling, bumping into, or quarreling while students are active in school. In the violence domain, experience of clinical injury treatment due to violence in the previous 12 months was selected. 

In the injury prevention domain, seatbelt usage in the front seat of a car, seatbelt usage in the rear seat of a car, and seatbelt usage on an express bus were divided into “never riding”, “always”, “usually”, “sometimes”, and “never.” Wearing helmets while riding a motorcycle/bicycle over the last 12 months was divided in the same way as seatbelt wearing. 

In the SSE domain, injury experience in schools, clinical treatment due to injuries at school, and experience of clinical injury treatment due to careless use of earphone/smartphone usage in the last 12 months were divided into “yes” or “no.” Safety education experience, injury prevention education, rescue and lifesaving, and CPR training were divided into “yes” or “no.” Danger evacuation training means the kinds of school safety education that included dangers such as earthquakes, fire and risk factors to life. In the violence domain, clinical injury treatment due to violence in the last 12 months was divided and analyzed into “none”, “1”, “2”, “3”, “4”, “5”, and “6 or more” (Table 1). 

### 2.3. Statistical Analysis

The data were analyzed by using IBM SPSS 25.0^®^ (IBM Corp., New York, NY, USA) to assess the relationship between TFE and explanatory variables by complex-samples chi-squared test and complex-samples multivariable logistic regression analysis.

In the sampling process, KYRBS stratified the population to minimize sampling errors, divided residence regions and schools into strata using stratification variables, and when there was only one colony in the strata, integrated it with the adjacent sample design layer. Since the integrated sample design strata is disclosed in the raw data, the sample design strata variable should be used as the stratification variable in data analysis [28]. Accordingly, in the KYRBS analysis, it was essential to use the analysis of complex samples represented by weights based on the population. Therefore, after the planning file was created using stratified variables (kstrata), colony variables (cluster), and weighted variables (w), complex sample analysis was performed. The results of the analysis are expressed as estimated percentages, odds ratios (ORs), and 95% confidence intervals (CIs), and the significance level was determined as type I error of 0.05.

## 3. Results

The TFE was 11.4% in all adolescents, 11.8% in males, and 11.1% in females, and increased with a higher grade (Table 2).

The TFE was 11.5% among adolescents who had never sat in the front seat of a car, 11.0% among those who always wore a seatbelt in the front seat of a car, and 13.8% among those who never wore a seatbelt in the front seat of a car. Thus, not wearing a seatbelt in the front seat of a car was associated with a higher TFE. Similar findings were obtained for not wearing a seatbelt in the rear seat of a car (11.0% among adolescents who had never sat in the rear seat of a car, 11.2% among those who always wore a seatbelt in the rear seat of a car, and 12.6% among those who never wore a seatbelt in the rear seat of a car), not wearing a helmet while riding a motorcycle (10.8% among adolescents who never rode a motorcycle, 15.4% among those who always wore a helmet when riding a motorcycle, and 18.3% among those who never wore a helmet while riding a motorcycle), and not wearing a helmet while riding a bicycle (10.6% among adolescents who never rode a bicycle, 12.7% among those who always wore a helmet while riding a bicycle, and 13.5% among those who never wore a helmet while riding a bicycle) (*p* < 0.001 for all comparisons). These findings are summarized in Table 3.

The TFE was 10.6% among adolescents who had never had an injury at school and 12.7% among those who had. Thus, TFE was higher in adolescents with injuries in schools than in adolescents without such injuries. Similarly, TFE was 10.9% among adolescents who had never received clinical treatment for an injury sustained at school in the last 12 months and 13.9% for those who had. Thus, the TFE of adolescents who had received clinical treatment was higher than that of those who had never experienced receiving such treatment. The TFE was 11.3% among adolescents who had never received clinical treatment due to earphone-/smartphone-related injury at school in the last 12 months and 21.0% among those who had received such treatment. Thus, the experience of clinical treatments for earphone-/smartphone-related injuries were associated with a higher TFE (*p* < 0.001, Table 4).

The TFE was 11.2% among adolescents who had received SSE in the last 12 months and 13.5% among those who had not, indicating that absence of SSE was associated with a higher TFE. Similar findings were obtained for danger evacuation education (11.1% for those who received danger evacuation education at school vs. 12.9% for those who did not), while the findings were reversed for rescue and lifesaving education (11.2% for those who did not receive rescue and lifesaving education at school vs. 11.9% for those who did). In contrast, TFE among adolescents who did not receive CPR training at school (11.4%) was not significantly different (*p* = 0.851) from that among adolescents who did receive CPR training at school (11.5%). These findings are summarized in Table 4.

The TFE was 11.3% among adolescents who had no experience of clinical treatment due to violence and 29.5% among those with six or more such clinical treatments. Thus, TFE was higher among adolescents with more clinical treatment experiences related to violence (*p* < 0.001, Table 5).

Table 6 shows the results of two predictive models for the detailed factors affecting TFE related to wearing a seatbelt or helmet in the last 12 months. In model one, adjusted for sex and grade, the factors influencing TFE were seatbelt wearing in the front seat of a car, seatbelt wearing in the rear seat of a car, seatbelt wearing on an express bus, helmet wearing while riding a motorcycle, and helmet wearing while riding a bicycle. In model two, adjusted for all related variables and grade, the factors influencing TFE were seatbelt wearing on an express bus, helmet wearing while riding a motorcycle, and helmet wearing while riding a bicycle (*p* < 0.05, Table 6). 

Table 7 presents two predictive models for the detailed factors affecting TFE in relation to clinical treatment experience due to injuries in the past 12 months. In model one, the factors affecting TFE were experience of clinical injury treatment, experience of clinical injury treatment associated with the use of earphones/smartphones, and experience of clinical injury treatment due to violence. In model two, the TFE was high in adolescents from higher grades and those with experience of injury in schools, clinical injury treatment, clinical injury treatment related to earphone/smartphone usage, and clinical treatment due to violence (*p* < 0.05, Table 7).

Table 8 presents two predictive models outlining the detailed factors affecting TFE related to school safety education in the past 12 months. In model one, the factors affecting TFE were safety education, danger evacuation education, and rescue and lifesaving education. In model two, the TFE was high in adolescents from higher grades, those who did not receive school safety education and danger evacuation education, and those who received rescue and lifesaving education (*p* < 0.05, Table 8).

## 4. Discussion

The overall rate of TFE in the last 12 months among all adolescents was 11.4%. In the National Oral Health Survey conducted by the Ministry of Health and Welfare in Korea in 2010, the dental trauma experience rate was 17.7% in 12 year-olds and 19.3% in 15 year-olds [3,31]. In 14–16 year-olds in Yangsan, Korea, the rate of dental trauma was reported to be 16.8% [5]. Thus, the rate of TFE in this study was not higher than those reported previously. 

There were no significant differences in TFE between males and females with regard to wearing seatbelts and helmets, clinical treatment experience due to injury, and SSE. These results are different from those reported by other studies in which males had higher rates of TFE than females [11,12,15]. This discrepancy may be attributed to the fact that this study was the result of a questionnaire analysis based on the participants’ experience over the past 12 months; however, further research may be warranted.

In the injury prevention domain, TFEs of adolescents not wearing a seatbelt on an express bus and not wearing a helmet when riding a motorcycle or bicycle were high (Table 6). The use of a seatbelt on a bus can significantly reduce the rate of head and chest injuries [32,33], while wearing a helmet is highly recommended to prevent head and neck injuries when riding a motorcycle or bicycle [34,35,36]. Helmets are recommended to prevent head and neck injuries caused by injuries involving electric scooters that have been gaining popularity recently [37]. Lieger et al. [7] reported that patients with a fracture of the mandible were most likely to have a dental injury. Facial injuries, including those regarding the mandibular jaw, can be directly related to tooth fracture. When using transportation, wearing a helmet, or wearing a seat belt can prevent damage to the head and face due to external damage, which means that tooth fracture can be prevented. 

The TFE among adolescents with injury treatment experience in schools was higher than that among adolescents without injury treatment experience, and adolescents who had injuries at school had higher TFEs than those without injuries at school (Table 7). In our multivariable logistic regression model, the adjusted OR for TFE was high among adolescents with experience of clinical injury treatment in schools. Thus, it appears that adolescents who experienced injuries at school appear to have had trauma to the maxillofacial area and teeth. The rate of injuries eligible for compensation by the National School Safety Mutual Service has continued to increase, and it is higher than the rate of occupational safety injuries compiled by the Ministry of Employment [38]. Since tooth damage continues to be recorded in injury claims filed with the School Safety Association [39], it is necessary to strengthen safety education for injury prevention in schools.

The adjusted ORs of TFE for clinical injury treatment related to earphone/smartphone usage were the highest among all the evaluated factors (Table 7). Trauma injuries involving the maxillofacial area and teeth have been increasing because of traffic injuries caused by the use of devices such as earphones and mobile phones, which are frequently used by adolescents. Kim [40] developed earphones that sound well at low volumes, which allow users to be more aware of their surroundings while walking with their earphones plugged in; however, it is fundamentally difficult to prevent distractions due to the use of earphones. Therefore, school safety education programs targeting adolescents should include aim to prevent adolescents from using these smart devices while walking inside or outside the school.

In the violence domain, the higher the number of clinical treatments due to violence (physical assault, intimidation, and bullying), the higher the TFE (Table 7). Lee [41] analyzed 343 cases of dental trauma in pediatric dentistry at a dental hospital in 2007–2009 and reported that 9.1% cases of permanent trauma to permanent teeth were attributable to fighting. Another study [42] analyzed 252 dental trauma patients who had been admitted to the hospital emergency room at night over two years and had received dental emergency treatment; 17.5% of the traumatic causes were due to fighting. Bae [43] analyzed 508 cases of 1–18 year-olds who were admitted to the hospital for dental trauma in 2009 and reported many cases of dental damage (2.76%) due to violence in this age group. Garbin et al. [44] reported that many tooth fractures were caused by domestic violence among adolescents aged 11–19 years. Ruslin et al. [45] reported that the most frequent causes of injuries in patients with maxillofacial fractures with dental trauma, who presented at the VU Medical Center in Amsterdam, were bicycle traffic injuries (39.0%), followed by falls (26.2%), and motorcycle traffic injuries (12.2%). Therefore, to prevent dental trauma, including tooth fractures, among adolescents, education to prevent traffic injuries and violence should be strengthened in SSE.

TFE was higher among adolescents without SSE than among those who received SSE in the past 12 months (Table 8). In accordance with the guidelines of the Ministry of Education, safety education in elementary and secondary schools in Korea is divided into seven major categories: 25 middle classes and 52 sub-classes. The seven major categories of safety education are life safety, traffic safety, violence and personal safety, drug and cyber addiction, disaster safety, occupational safety, and first aid [46,47]. Among the topics of safety education in middle and high schools, the topics that are supposed to be related to the protection of tooth fractures are school violence prevention, pedestrian safety, bicycle safety, motorcycle safety, and public transportation safety [47]. To prevent injuries, including tooth fractures, in adolescents, it is necessary to strengthen SSE.

Danger evacuation and rescue and lifesaving education are detailed SSE topics that showed significant differences in TFE depending on the presence or absence of safety education (Table 8). Danger evacuation education is one of the seven safety education areas in the disaster safety area, and fire, social disaster, and natural disaster are classified in the middle category [46]. There are few existing studies on the actual contribution of disaster safety education to the protection of tooth or body damage. However, disasters can lead to personal and physical damage, so it is necessary to strengthen them through SSE [48].

Meanwhile, unlike other results, the TFE of adolescents who had rescue and lifesaving education was higher than those who did not have this education (Table 8). Rescue and lifesaving education is not included in the seven areas of SSE, and as part of the experience education, it is often conducted as a preparatory stage before on-site experience education such as water rafting or marine sports such as skin scuba. Since rescue life-saving education is provided before and after radical experiential learning, adolescents who received this education may have suffered many injuries and physical injuries [49,50]. Therefore, more detailed consideration regarding educational supplementation is necessary for administering rescue life-saving education before any field experience education in areas such as marine sports.

Thus, overall, the risk factors related to TFE in Korean adolescents were health-risk behaviors such as not wearing seatbelts on an express bus, not wearing helmets while riding motorcycles and bicycles, clinical treatment experiences due to injuries at school, usage of earphones or smartphones, and lack of experience with SSE such as danger evacuation protocols. The top risk factor related to tooth fractures was injuries associated with the use of mobile devices (earphones or smartphones), followed by inexperience of SSE.

We also concluded that it was necessary to strengthen safety education for the use of smart devices such as earphones and mobile phones and to improve the school environment to prevent TFE due to facial injuries in school. This study only analyzed the questionnaires related to the variables that caused TFE. Environmental factors, including those outside of school and home, were not reviewed. It was not possible to review the factors encountered in external activities and on-the-job training, which adolescents frequently engage in, and further studies are warranted to address this aspect. Nevertheless, the results of this study analyze the factors associated with TFE among adolescents by using national representative data and may contribute to design interventions to prevent tooth fractures.

## 5. Conclusions

The most relevant factor related to TFE was injuries caused by the usage of mobile devices, followed by lack of SSE. It is necessary to improve the school environment so that tooth fractures do not occur due to facial injuries in school. Moreover, safety education should include education regarding the use of smart devices such as earphones and mobile phones in school and wearing a seatbelt on an express bus and wearing a helmet when riding a motorcycle or bicycle. In addition, wearing a mouthguard should be recommended when performing vigorous exercise with high levels of physical contact. Adolescents who received school safety education are highly conscious of safety, and those with a high level of safety consciousness have a lower risk of injury, which is explained in connection with tooth fracture and injury.

## Figures and Tables

**Table 1 ijerph-17-08556-t001:** Variables for injury prevention, school safety education, and violence related to tooth fracture experience from the 14th Korea Youth Risk Behavior Survey in 2018.

Domain	Duration	Variables
Injury prevention	Usually	Seatbelt wearing in car front seat/car rear seat
	Past 12 months	Helmet wearing while riding motorcycle/bicycle, experience of clinical injury treatment, experience of clinical injury treatment due to careless use of earphones/cellphones.
School safety education (SSE)	Past 12 months	School education experience of safety, injury prevention, danger evacuation, rescue and lifesaving, CPR training *
Violence	Past 12 months	Experience of clinical injury treatment due to violence

* Cardiopulmonary resuscitation training using mannequins.

**Table 2 ijerph-17-08556-t002:** Tooth fracture experience in the past 12 months.

Grade	Total		Male		Female		*p*-Value *
	*N*	%	*N*	%	*N*	%	
All	60,040	11.4	30,463	11.8	29,577	11.1	0.023
Middle school							
Grade 1	9847	11.0	4960	11.8	4887	10.1	0.011
Grade 2	10,092	11.2	5137	11.4	4955	10.9	0.505
Grade 3	10,290	11.5	5231	11.3	5059	11.7	0.611
High school							
Grade 1	9260	11.4	4805	12.0	4455	10.8	0.080
Grade 2	10,039	10.9	5110	11.2	4929	10.5	0.251
Grade 3	10,512	12.5	5220	12.7	5292	12.4	0.626
*p*-Value ***		0.009		0.207		0.007	

* Complex-samples chi-squared test; *N*: unweighted value, %: tooth fracture experience, weighted value; age of adolescents: grade 1 of middle school, 12 years; grade 3 of high school, 15 years.

**Table 3 ijerph-17-08556-t003:** Tooth fracture experience in the past 12 months by seatbelt or helmet wearing.

Variables	Wearing Frequency (%)					*p*-Value *
	Never Riding	Always	Usually	Sometimes	Never	
Seatbelt wearing in car front seat	11.5	11.0	12.0	11.8	13.8	<0.001
Seatbelt wearing in car rear seat	11.0	11.2	10.5	10.9	12.6	<0.001
Seatbelt wearing in express bus	10.6	10.8	11.4	13.2	13.4	<0.001
Helmet wearing while riding motorcycle	10.8	15.4	14.3	18.2	18.3	<0.001
Helmet wearing while riding bicycle	10.6	12.7	11.6	11.2	13.5	<0.001

The values mean percentage of tooth fracture experience, which means symptoms of the tooth being fractured or broken, or that the tooth was fractured or broken due to movement or injury during the last 12 months. * Complex-samples chi-square test.

**Table 4 ijerph-17-08556-t004:** Tooth fracture experience in the past 12 months by clinical treatment due to injury in school.

Variables	Contents	Tooth Fracture (%)		*p*-Value *
		Inexperience	Experience	
Clinical treatment experience	Injury in school	10.6	12.7	<0.001
	Clinical treatment for injury at school	10.9	13.9	<0.001
	Clinical injury treatment due to careless use of earphones/smartphones	11.3	21.0	<0.001
School safety education	Safety	13.5	11.2	<0.001
	Injury prevention	11.9	11.2	0.014
	Danger evacuation	12.9	11.1	<0.001
	Rescue and lifesaving	11.2	11.9	0.009
	CPR training ^†^	11.4	11.5	0.851

The values mean percentage of tooth fracture experience, which means symptoms of the tooth being fractured or broken, or that the tooth was fractured or broken due to movement or injury during the last 12 months. * Complex samples chi-square test; ^†^ cardiopulmonary resuscitation training using mannequins.

**Table 5 ijerph-17-08556-t005:** Tooth fracture experience by frequency of clinical injury treatments due to violence in the past 12 months.

Variable	Frequency of Treatments Due to Violence							*p* *
	None	1	2	3	4	5	6	
Tooth fracture experience (%)	11.3	20.3	12.2	11.8	20.2	18.2	29.5	0.001

* Complex samples chi-square test.

**Table 6 ijerph-17-08556-t006:** The adjusted association of the tooth fracture experience in the past 12 months with variables related to seatbelt or helmet usage assessed by two complex-sample multivariable logistic regression models.

Variables	Model 1	Model 2
	Adjusted OR (95% CI)	Adjusted OR (95% CI)
Sex (Ref. = Female)		0.99 (0.94–1.05)
Grade		1.02 (1.00–1.04)
Seatbelt wearing in car front seat (Ref. = No riding or always wearing)	1.06 (1.03–1.09)	1.01 (0.98–1.04)
Seatbelt wearing in car rear seat (Ref. = No riding or always wearing)	1.05 (1.03–1.07)	1.01 (0.99–1.04)
Seatbelt wearing in express bus (Ref. = No riding or always wearing)	1.08 (1.06–1.11)	1.06 (1.04–1.08)
Helmet wearing while riding motorcycle * (Ref. = No riding or always wearing)	1.17 (1.15–1.20)	1.14 (1.11–1.17)
Helmet wearing while riding bicycle * (Ref. = No riding or always wearing)	1.07 (1.05–1.08)	1.05 (1.03–1.06)

* In the past 12 months; dependent variable: tooth fracture experience (reference category = inexperience); model 1: adjusted for sex and grade; model 2: adjusted for sex, grade, seatbelt wearing in the front seat of a car, seatbelt wearing in the rear seat of a car, seatbelt wearing on express bus, helmet wearing while riding motorcycle, and helmet wearing while riding bicycle.

**Table 7 ijerph-17-08556-t007:** The adjusted association of tooth fracture experience in the past 12 months with variables related to clinical treatment experience due to injury by a complex-sample multivariable logistic regression model.

Variables	Model 1	Model 2
	Adjusted OR (95% CI)	Adjusted OR (95% CI)
Sex (Ref. = Female)		1.03 (0.98–1.10)
Grade		1.03 (1.01–1.04)
Experience of injury in school (Ref. = No)	1.23 (1.17–1.30)	1.13 (1.06–1.19)
Experience of clinical injury treatment in school (Ref. = No)	1.32 (1.24–1.41)	1.20 (1.11–1.29)
Experience of clinical injury treatment related to earphone/smartphone use (Ref. = No)	2.08 (1.76–2.45)	1.65 (1.38–1.96)
Experience of clinical injury treatment due to violence (Ref. = No)	1.19 (1.14–1.24)	1.13 (1.08–1.18)

Dependent variable: tooth fracture experience (reference category = inexperience); model 1: adjusted for sex and grade; model 2: adjusted for sex, grade, experience of injury in school, experience of clinical injury treatment in school, experience of clinical injury treatment related to earphone/smartphone use, experience of clinical injury treatment due to violence.

**Table 8 ijerph-17-08556-t008:** The adjusted association of tooth fracture experience in the past 12 months with variables related to safety education experience in school by a complex-sample multivariable logistic regression model.

Variables	Model 1	Model 2
	Adjusted OR (95% CI)	Adjusted OR (95% CI)
Sex (Ref. = Female)		1.04 (0.98–1.10)
Grade		1.02 (1.00–1.04)
Safety education (Ref. = Yes)	1.22 (1.13–1.31)	1.15 (1.04–1.28)
Injury prevention education (Ref. = Yes)	1.05 (1.00–1.11)	1.00 (0.94–1.06)
Danger evacuation education (Ref. = Yes)	1.17 (1.11–1.24)	1.14 (1.05–1.24)
Rescue and lifesaving education (Ref. = Yes)	0.93 (0.88–0.98)	0.88 (0.83–0.94)
CPR training ^†^ (Ref. = Yes)	0.98 (0.93–1.04)	0.97 (0.92–1.03)

^†^ Cardiopulmonary resuscitation training using mannequins; dependent variable: tooth fracture experience (reference category = inexperience); model 1: adjusted for sex and grade; model 2: adjusted for sex, grade, safety education, injury prevention education, evacuation education, rescue and lifesaving education, and CPR training.

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
