# Peer review of "Injury Prevention, Safety Education and Violence in Relation to the Risk of Tooth Fracture among Korean Adolescents"

_ijerph, 2020, doi:10.3390/ijerph17228556_

Round 1

Reviewer 1 Report

Title. The title appears quite confusing. If “violence” is a negative word and can be associated to the term “risk”, “injury prevention” and “safety education” have a positive meaning. I would suggest the Authors to change it this way: “ Injury prevention, safety education and violence in relation to the risk of tooth fracture among Korean adolescents.” Furthermore, I would suggest to specify that the injury prevention is related to the use of protection tools, since the difference between the expression “injury prevention” and “safety education” does not appear evident.

Abstract. Line 22: why the Authors use the term “prevalence” rather than “incidence”? Please specify this.

             Line 24: “clinical treatment experiences due to accident injuries” is an     unclear expression; why an experience of clinical treatment for various accidents should  have been related to tooth fractures? I would suggest the Authors to use another term or to better explain what they want to say.

              Line 25: which kind of danger are the Authors referring to? A general danger? A fire danger? Please specify and put this datum in relation to the tooth fracture.  

Introduction. Lines 34-35: the Authors say “studies” but there is only a citation at the end of the sentence. Maybe this is the citation of citations. Anyway, the cited paper is too old. Please report updated data with the correct sources.

Line 40: the Authors report citations  about the maxillary central incisors: why from now on They speak  in general of TFE, tooth fracture experience, without focusing, for example, on the upper frontal teeth?  I think that it should have been specified which teeth the survey is referring to, since the fracture of frontal teeth is completely different as regards the causes and the consequences than that of posterior teeth. If the Authors did not focus on a particular tooth fracture, They should specify why they have taken into consideration all the teeth.  

Materials & Methods. Line 58: The Author should specify better some points about the  study design: this survey seems to be a single cohort retrospective study, with the aim to identify some prognostic predictor variables to be correlated with the event “tooth fracture”. Why did the Authors not consider a double parallel cohort study correlating the independent variables with the dependent variables of tooth fracture and no-tooth fracture?

Line 84: it looks unclear what exactly the Authors have taken from the KYRBS. It seems that They have acquired only the names of the subjects to be interviewed: in this case, why the survey is limited  to the previous 12 months? Moreover, If the Authors have taken from the KYRBS  cases of people who experienced a tooth fracture in the 12 months previous the KYRBS survey of 2018, it should  be clarified which year  the results obtained are referring to.   

Discussion.  Lines 234-236: why the survey was limited to the past 12 months? “Past” in respect to what? Furthermore, why a 12 month's survey should be different from other surveys over different periods?

Lines 310-312:  why the environmental factors have not been included? I would suggest the Authors to better underline that all the survey is related to the school environment, explaining why They have considered only these data. This appears a relevant factor.

Final comment: the article  analyzes in an organic way the risks of dental fractures in adolescents. Although well-structured, some doubts persist about the study design and the data collection; a few points should be clarified.

Author Response

Dear Reviewer

In fact, the reviewer gave me a great opinion. However, since there were many related opinions in the design and survey items of this study, it was not correct 100% as follow your opinions. However, the opinions you have given are valuable in my next research and have been a great help in improving my research skills. Thank you for your understanding.

Best regards,

Jin-Bom Kim

Reviewer 1

Final comment: the article analyzes in an organic way the risks of dental fractures in adolescents. Although well-structured, some doubts persist about the study design and the data collection; a few points should be clarified.

  1. The title appears quite confusing. If “violence” is a negative word and can be associated to the term “risk”, “injury prevention” and “safety education” have a positive meaning. I would suggest the Authors to change it this way: “ Injury prevention, safety education and violence in relation to the risk of tooth fracture among Korean adolescents.” Furthermore, I would suggest to specify that the injury prevention is related to the use of protection tools, since the difference between the expression “injury prevention” and “safety education” does not appear evident.
  • Thank you for your suggestion, we changed the title in to Injury prevention, safety education and violence in relation to the risk of tooth fracture among Korean adolescents.

  1. Line 22: why the Authors use the term “prevalence” rather than “incidence”? Please specify this.
  • Incidence proportion is the proportion of an initially disease-free population that develops disease, becomes injured, or dies during a specified (usually limited) period of time. Prevalence, sometimes referred to as prevalence rate, is the proportion of persons in a population who have a particular disease or attribute at a specified point in time or over a specified period of time. Prevalence differs from incidence in that prevalence includes all cases, both new and preexisting, in the population at the specified time, whereas incidence is limited to new cases only. We can not follow up and count new patients for tooth fracture. Therefore, we used the terms “prevalence”.

[Reference] Last JM. A dictionary of epidemiology, 4th ed. New York: Oxford U. Press; 2001.

  1.  Line 24: “clinical treatment experiences due to accident injuries” is an     unclear expression; why an experience of clinical treatment for various accidents should  have been related to tooth fractures? I would suggest the Authors to use another term or to better explain what they want to say.

  • Thank you for good suggestion. It is judged that the meaning of the term presented in the questionnaire is unclear in the process of using it by translating it into English without modifications. The terms mean that students have been injured in school (within school boundaries such as classrooms, hallways, playgrounds, etc.) and have been treated at a hospital. It refers to a situation caused by falling, bumping into, or quarreling while students are active in school.

We changed the term in to “clinical treatment due to accident injuries at school”

  1. Line 25: which kind of danger are the Authors referring to? A general danger? A fire danger? Please specify and put this datum in relation to the tooth fracture.

  • Thank you for good suggestion. Danger evacuation training means the kinds of school safety education included dangers such as earthquake, fire and risk factors in life. In this study, adolescents who received school safety education sincerely are highly conscious of safety, and adolescents with high safety consciousness have a lower risk of injury, which is explained in connection with tooth fracture and injury. Because significant variables were briefly presented in the abstract, it is judged that the detailed description of the danger was insufficient. I will supplement this in detail in the text. Please check out in Line 125-127 and Line 330-332.

  1. Lines 34-35: the Authors say “studies” but there is only a citation at the end of the sentence. Maybe this is the citation of citations. Anyway, the cited paper is too old. Please report updated data with the correct sources.

à Thank you for good suggestion. We changed old reference into new one.

[1] Petti, S.; Glendor, U.; Andersson, L. World traumatic dental injury prevalence and incidence, a meta-analysis—One billion living people have had traumatic dental injuries. Dent. Traumatol. 2018, 34, 71–86.

  1. Line 40: the Authors report citations  about the maxillary central incisors: why from now on They speak  in general of TFE, tooth fracture experience, without focusing, for example, on the upper frontal teeth?  I think that it should have been specified which teeth the survey is referring to, since the fracture of frontal teeth is completely different as regards the causes and the consequences than that of posterior teeth. If the Authors did not focus on a particular tooth fracture, They should specify why they have taken into consideration all the teeth.  

  • Unfortunately, in this study, only tooth fracture was investigated through self-answering a questionnaire. Therefore, we could not pay attention to the detailed tooth location. The risk behavior of adolescents is compared with whether or not they have fractured teeth in the form of a self-reported questionnaire, not data obtained through actual oral examination. We will analyze the tooth fracture location in the oral examination data later.

  1. Materials & Methods.Line 58: The Author should specify better some points about the  study design: this survey seems to be a single cohort retrospective study, with the aim to identify some prognostic predictor variables to be correlated with the event “tooth fracture”. Why did the Authors not consider a double parallel cohort study correlating the independent variables with the dependent variables of tooth fracture and no-tooth fracture?
  • The reviewer's comments suggest ways to improve the quality of the research. Unfortunately, this study is a cross sectional design, not a cohort design. In this study, a logistic regression analysis was performed to determine whether a tooth fractured. The researcher acknowledges that there are limitations in the analysis, and in future studies, subjects without tooth fractures will be compared as a control group.

  1. Line 84: it looks unclear what exactly the Authors have taken from the KYRBS. It seems that They have acquired only the names of the subjects to be interviewed: in this case, why the survey is limited to the previous 12 months? Moreover, If the Authors have taken from the KYRBS  cases of people who experienced a tooth fracture in the 12 months previous the KYRBS survey of 2018, it should  be clarified which year  the results obtained are referring to.   

  • The reviewer's comments suggest ways to improve the quality of the research. Thank you. However, a single-year survey was conducted in this KYRBS, and matching with previous studies did not allow a long period of time to be seen. Since the individual code of the subject who responded previously was not presented, this study is a cross sectional design, not a cohort design. Until now, research using KYRBS has been conducted for a single year, so if we find a way to match the response code of individual subjects, we will try it in future research using the method suggested by the reviewer. Thank you.

  1.  Lines 234-236: why the survey was limited to the past 12 months? “Past” in respect to what? Furthermore, why a 12 month's survey should be different from other surveys over different periods?
  • Most of the questionnaire questions ask about the health risk activities of youth in the past 12 months. That's what 12 months means. As we mentioned above in 10, research using KYRBS has been conducted for a single year, so if we find a way to match the response code of individual subjects, we will try it in future research using the method suggested by the reviewer. Thank you.
  1. Lines 310-312:  why the environmental factors have not been included? I would suggest the Authors to better underline that all the survey is related to the school environment, explaining why They have considered only these data. This appears a relevant factor.
  • The research item did not ask if the environment around the subject was safe. Therefore, I will describe in addition to the fact that environmental factors cannot be accurately grasped due to the limitations of this study.

Reviewer 2 Report

Dear Authors

the article is interesting. By the way, the manuscript has many flaws,it must be improved taking into consideration some suggestions I attached.

Best regards

Author Response

Reviewer 2

We sincerely thank the reviewers for their careful consideration and valuable comments. We have provided a point-by-point response to each of the reviewers’ comments below. We sincerely look forward to a positive response regarding the publication of our manuscript.

Best regards,

Jin-Bom Kim

ABSTRACT

  1. The subtitles are not not present: aim, materials and methods, results and conclusions.
  • In accordance with the submission guidelines, it is required to write without subtitles.

  1. According to the article: Thus, to prevent tooth fractures among adolescents, schools should strengthen but this part of the abstract takes into consideration only one study variable (safety education) without any conclusion respect to the other variables (injury prevention and violence).
  • The reviewer's comments suggest ways to improve the quality of the research. Thank you. We added a sentence in conclusion as below.

: “Care should be taken to manage facilities around the school and to prevent injury related to tooth fracture.”

INTRODUCTION

  1. According to the article: The hypotheses of this study were as follows: (1) Adolescents riding a motorcycle or bicycle without a helmet will have a higher TFE than those who use a helmet. 2) Adolescents who did not receive SSE will but why consider hypotheses that are not part of the determined aims and also that in the conclusions part of this article they are not mentioned?
  • Thank you. We mentioned emphasis for wearing a seatbelt in an express bus and wearing a helmet when riding a motorcycle or bicycle. Please confirm Lines 29 and 330-331.

: Moreover, safety education should include education regarding the use of smart devices such as earphones and mobile phones in schools and wearing a seatbelt in an express bus and wearing a helmet when riding a motorcycle or bicycle.

MATERIALS AND METHODS

  1. The sample is not homogeneous, and there is no reference to the age range of the participants, which would be adequate to form a more homogeneous sample and greatly reduce the biases in the results obtained.

-à Thank you for good suggestion. This study was conducted for students enrolled in the first year of middle school to the third year of high school as a result of a questionnaire that students responded by themselves at school. We mentioned the ages of participants Line in 96.

: The participations were aged 13- to 18- year-old adolescence.

  1. According to the article: In the injury prevention domain, seatbelt usage in the car front seat, seatbelt usage in the car rear but in table 1, the domain accident injury should be changed to injury preventions as established in the aim of the present study.
  • Thank you for good suggestion. As your suggestion, we changed it into “Injury prevention” in Table 1.

RESULTS

  1. The tables 2, 3, 5, 6 and 7 are more descriptive than analytical, the statistical results obtained should be better interpreted to obtain a correct conclusion. According to the article: The TFE was 11.3% among adolescents who had no experience of clinical treatment due to violence and 29.5% among those with 6 or more such clinical treatment experiences. Thus, TFE was higher among adolescents with more clinical treatment experiences related to violence (p < 0.001, Related table was not shown). this reason the results obtained from this variable help to obtain a precise conclusion, for this reason its results should be shown in a statistically analysed table.
  • Thank you. We added related table as Table 5. And then we re-numbered the table numbers.

Table 5. Tooth fracture experience by frequency of clinical injury treatments due to violence in the past 12 months

Variable

Frequency of treatments due to violence

P*

None

1

2

3

4

5

6

Tooth fracture experience (%)

11.3

20.3

12.2

11.8

20.2

18.2

29.5

0.001

*Complex samples chi-square test

DISCUSSION

According to the article: The TFE among adolescents with injury treatment experience in schools was higher than that among adolescents without injury treatment experience, and adolescents who had injuries at school had higher TFEs than those without injuries at school. In our multivariable logistic regression model, the adjusted OR for TFE was high among adolescents with experience of clinical injury treatment in but in this part of the article, it would be advisable to place the number of the results table in parentheses to compare the textual interpretation of the discussion with the numerical interpretation of the results in this scientific article.

à Thank you. We added table numbers into discussion parts.

CONCLUSION

The conclusion should be more focused on determining what is the relationship between the dependent variable TFE (tooth fracture experience) and the 3 independent variables (injury prevention, safety education and violence). It is necessary to interpret the results obtained with greater precision and carry out a statistical analysis that responds to the aim determined in this scientific investigation.

  • Thank you. We supplemented the conclusion parts.

: The most relevant factor related to TFE was accident injuries caused by the usage of mobile devices, followed by lacks of SSE. It is necessary to improve the school environment such that tooth fractures do not occur due to facial injuries in school. Moreover, safety education should include education regarding the use of smart devices such as earphones and mobile phones in schools and wearing a seatbelt in an express bus and wearing a helmet when riding a motorcycle or bicycle. In addition, wearing a mouthguard should be recommended when performing vigorous exercises with high levels of physical contact. Adolescents who received school safety education sincerely are highly conscious of safety, and they with high safety consciousness have a lower risk of injury, which is explained in connection with tooth fracture and injury.

Reviewer 3 Report

Comments

The article although not a noble one, still shades light on some newer aspects such as, the link between careless use of headphones and/smartphones and traumatic dental injuries in adolescents, which makes it a good read. It is relevant with current time and social scenario. The article can lead to further research regarding updating and upgrading School Safety Education. The manuscript could use some improvement as per suggestion below,

  1. The title needs revising. Does not make proper sense. Can be “Role of accidental injury, lack of safety education and violence related risk factors in occurrence of tooth fracture.”

  1. Should define tooth fracture. Which criteria included.

  1. Line 23: Should be “not wearing helmet while riding motorcycle” not “in motorcycle”.

  1. Keywords preferably, should be limited within 4 to 5 words.

  1. Line 34-35: Should use a newer reference in place of reference 1, it is outdated.

  1. Line 68: Violence is a risk factor for dental injury. However “injury prevention and safety education” are not risk factors but are measures taken for safety. This confusion in terminology has been observed though out the manuscript. Please revise.

  1. Line 97: Since the questionnaire is the main tool for observation of all variables, references should be added to how they influenced the preparation of a rational one. The model for the questionnaire can be attached to the back matter as well. I will facilitate future researchers in conducting similar studies.

  1. Line 109-110: “experience of clinical injury treatment”- Not clear which injuries are included in this category. Mentioned “experience of clinical injury treatment during earphone/smartphone usage”- should rather be “experience of clinical injury treatment due to careless use of earphone/smartphone.

  1. Table 2 and Table 3: The percentage calculations in relation to the “N” shown are not clear.

  1. Table 4: “Experience” and “Inexperience” mentioned under Tooth Fracture does not state the trait clearly. The term “SSE” should be mentioned for clarification.

  1. Results of the study related to tooth fracture due to not using helmets and sit belts should be discussed more in the Discussion section as it is an important part of the hypotheses. Related reference should be used for comparison if available.

  1. Line 317-322: These lines in the Conclusion are word to word duplicates of line 22-27 of the Abstract. If the same thing needs to be repeated it should be narrated in a different manner avoiding duplication.

Author Response

Reviewer 3

We sincerely thank the reviewers for their careful consideration and valuable comments. We have provided a point-by-point response to each of the reviewers’ comments below. We sincerely look forward to a positive response regarding the publication of our manuscript.

Best regards,

Jin-Bom Kim

The article although not a noble one, still shades light on some newer aspects such as, the link between careless use of headphones and/smartphones and traumatic dental injuries in adolescents, which makes it a good read. It is relevant with current time and social scenario. The article can lead to further research regarding updating and upgrading School Safety Education. The manuscript could use some improvement as per suggestion below,

  1. The title needs revising. Does not make proper sense. Can be “Role of accidental injury, lack of safety education and violence related risk factors in occurrence of tooth fracture.”
  • Thank you. We discussed for changing the title, and then we changed it as below.

 : Injury prevention, safety education and violence in relation to the risk of tooth fracture among Korean adolescents

  1. Should define tooth fracture. Which criteria included.
  • Since the data was not obtained through oral examination, we can respond as follows. Data obtained by self-response questionnaire, inclusion criteria based on clinical evaluation criteria were not presented in the data. The questionnaire received in the actual response from the survey is detailed.

: Tooth fracture was included if the participants answered that they experienced symptoms of the tooth fractured or broken, or that the tooth was fractured or broken due to movement or accident during the last 12 months.

  1. Line 23: Should be “not wearing helmet while riding motorcycle” not “in motorcycle”.
  • Thank you. We corrected 12 times in the whole text.

  1. Keywords preferably, should be limited within 4 to 5 words.
  • We deleted 2 key words, which were not relatively important.  

: adolescent, health behavior, injury prevention, oral health, safety education, tooth fracture, violence

  1. Line 34-35: Should use a newer reference in place of reference 1, it is outdated.

à Thank you. We changed it as below. In Line 33-34.

: The prevalence of traumatic dental injuries frequency of primary and permanent dentition was 22.7% and 15.2%, respectively [1].

 [Ref. 1] Petti, S.; Glendor, U.; Andersson, L. World traumatic dental injury prevalence and incidence, a meta-analysis—One billion living people have had traumatic dental injuries. Dent. Traumatol. 2018, 34, 71–86.

  1. Line 68: Violence is a risk factor for dental injury. However “injury prevention and safety education” are not risk factors but are measures taken for safety. This confusion in terminology has been observed though out the manuscript. Please revise.

  • Thanks for your suggestion. We corrected it as below. In Line 67-69

 : Therefore, this study aimed to analyze prevention or risk factors such as injury prevention, safety education, and violence in relation to tooth fractures among adolescents by using the 14th Korea Youth Risk Behavior Survey (KYRBS) data obtained from middle and high school students.

  1. Line 97: Since the questionnaire is the main tool for observation of all variables, references should be added to how they influenced the preparation of a rational one. The model for the questionnaire can be attached to the back matter as well. I will facilitate future researchers in conducting similar studies.
  • Thanks for your suggestion. The questionnaire is written in Korean and opened to public freely. So researcher can download the questionnaire in freely. Looking for references on questionnaire development, it was confirmed that a report on the development of a youth health risk behavior survey system in 2005 had an expert advisory meeting for questionnaire development. References or specific data presented at the advisory meeting could not be confirmed. As detailed conference data are not provided, the report is presented as a reference.

[Added reference]

[30] Ministry of Education; Korea Centers for Disease Control and Prevention Development of Korea Youth Risk Behavior Survey system in 2005. Korea Centers for Disease Control and Prevention. 2005, Seoul, Korea, 59-300.

  1. Line 109-110: “Experience in hospital treatment for injury in school”- Not clear which injuries are included in this category. Mentioned “experience of clinical injury treatment during earphone/smartphone usage”- should rather be “experience of clinical injury treatment due to careless use of earphone/smartphone.

à Thanks for your suggestion. We changed the “experience of clinical injury treatment during earphone/smartphone usage” into “experience of clinical injury treatment due to careless use of earphone/smartphone. And we added the explanations for “Experience in hospital treatment for injury in school”.

: The experience of treatment due to injury at school means that students have been injured in school (within school boundaries such as classrooms, hallways, playgrounds, etc.) and have been treated at a hospital. It refers to a situation caused by falling, bumping into, or quarreling while students are active in school.

  1. Table 2 and Table 3: The percentage calculations in relation to the “N” shown are not clear.
  • Thanks for your suggestion. We corrected it as below.

: [Table 2] %: tooth fracture experience, weighted value

: [Table 3] The values mean percentage of tooth fracture experience.

  1. Table 4: “Experience” and “Inexperience” mentioned under Tooth Fracture does not state the trait clearly. The term “SSE” should be mentioned for clarification.
  • Thanks for your suggestion. We added sentences for tooth fracture under the Table 3 and 4 as below. The term “SSE” is explained in the text Line 122-125.

: The values mean percentage of tooth fracture experience, which means symptoms of the tooth fractured or broken, or that the tooth was fractured or broken due to movement or accident during the last 12 months.

  1. Results of the study related to tooth fracture due to not using helmets and sit belts should be discussed more in the Discussion section as it is an important part of the hypotheses. Related reference should be used for comparison if available.
  • We mentioned more information and opinions for not using helmets and sit belts, Line 246-250.

 : Lieger et al [7] reported that patients with a fracture of the mandible were most likely to have a dental injury. Facial injuries, including the mandibular jaw, can be directly related to tooth fracture. When using transportation, wearing a helmet or wearing a seat belt can prevent damage to the head and face due to external damage, which means that it can be prevented in connection with tooth fracture.

  1. Line 317-322: These lines in the Conclusion are word to word duplicates of line 22-27 of the Abstract. If the same thing needs to be repeated it should be narrated in a different manner avoiding duplication.
  • Thank you for good point out. We deleted the duplication parts in conclusion as below in Line 325-326.

: The most relevant factor related to TFE was accident injuries caused by the usage of mobile devices, followed by lacks of SSE.

Round 2

Reviewer 2 Report

Dear Authors 

All the required modifications have been addressed and now the article is suitable for publication. 

Kind regards 

Author Response

Dear Reviewer

Thanks for answering.

Best regards,

Jin-Bom Kim

Reviewer 3 Report

Dear Authors, I’ve extensively read the present manuscript and I believe that, the manuscript is now suitable for publication.

Author Response

(The authors gave the same response as above.)
